# Association between Chronotype and Nutritional, Clinical and Sociobehavioral Characteristics of Adults Assisted by a Public Health Care System in Brazil

**DOI:** 10.3390/nu13072260

**Published:** 2021-06-30

**Authors:** Juliana C. Reis-Canaan, Marcelo M. Canaan, Patrícia D. Costa, Tamires P. Rodrigues-Juliatte, Michel C. A. Pereira, Paula M. Castelo, Vanessa Pardi, Ramiro M. Murata, Luciano J. Pereira

**Affiliations:** 1Health Sciences Faculty, Universidade Federal de Lavras (UFLA), Lavras 37200-900, MG, Brazil; reisjuliana@yahoo.com.br (J.C.R.-C.); marcelo.canaan@ufla.br (M.M.C.); patriciadaniela.costa@yahoo.com.br (P.D.C.); tamires.rodrigues@hotmail.com (T.P.R.-J.); deangelis@ufla.br (M.C.A.P.); 2Department of Pharmaceutical Sciences, Universidade Federal de São Paulo (UNIFESP), Diadema 09913-030, SP, Brazil; paula.castelo@unifesp.br; 3Department of Foundational Sciences, School of Dental Medicine, East Carolina University (ECU), Greenville, NC 27834, USA; pardiv19@ecu.edu

**Keywords:** diet, circadian rhythm, heart disease, micronutrients

## Abstract

Chronotype (CT) has been associated with predisposition to chronic noncommunicable diseases (CNCDs), such as diabetes mellitus and obesity. However, the effects of CT on individuals assisted by public health systems (PHSs) in middle-up economies are still poorly explored. The objective of this study was to evaluate the relationship between CT and clinical, sociobehavioral and nutritional aspects in adults assisted by a PHS in Brazil. This is a population-based cross-sectional study. The sample consisted of 380 individuals, selected through probabilistic sampling by clusters, in all health units in a city of approximately 100 thousand inhabitants. Data collection was performed during home visits, by means of general and nutritional interviews, anthropometric measurements and the Morningness–Eveningness Questionnaire (MEQ). Statistical analysis comprised chi-square test and principal component analysis (CPA) followed by Fisher’s discriminant analysis to determine aspects associated with each CT (morning, evening or intermediate). With the aim of explaining the variation in the CT scores, the consumption of micronutrients (corrected to the total energy intake) and other individual and sociodemographic variables were used as explanatory factors in the adjustment of a linear regression model. The morning group was characterized by older men, with less than eight years of schooling, with low body mass index (BMI) and with low intake of omega-6, omega-3, sodium, zinc, thiamine, pyridoxine and niacin. The evening group, on the other hand, was composed of younger individuals, with a high consumption of these same nutrients, with high BMI and a higher frequency of heart diseases (*p* < 0.05). It was concluded that most morning CT individuals were elderly thin males with lower consumption of omega-6 and -3, sodium, zinc, thiamine, pyridoxine and niacin, whereas evening individuals were younger, had higher BMI and had higher consumption of the studied micronutrients. The identification of circadian and behavioral risk groups can help to provide preventive and multidisciplinary health promotion measures.

## 1. Introduction

Chronotype (CT) represents an individual’s circadian phenotype (e.g., one’s behavioral preference) [1,2,3], generating performance patterns as morning, evening or intermediate types [4,5]. Although CT has a genetic basis, it is also influenced by environmental, biological and social factors [1,6,7,8]. Epidemiological studies demonstrate a normal distribution of CT in the population [3,9] and several tools may be used to determine CTs, ranging from questionnaires to hormonal measures [5,10,11].

Human beings are able to subvert the light–dark cycle, and metabolic functions together with social interactions may influence the circadian rhythm. Nutrient intake and meal distribution along the day/night periods and hormonal secretions integrate metabolism signals to the central clock [7,12]. The unbalance of the physiological synchrony may be associated with the appearance of chronic noncommunicable diseases (CNCDs) [1,13,14].

CNCDs such as diabetes mellitus and obesity represent important public health challenges worldwide due to the morbidity, mortality and elevated associated costs [15,16,17,18,19]. Studies have established a potential association between CT and the aforementioned diseases, mainly with regards to the eveningness preference [20,21]. Besides, a link between cancer and CT has also been reported [13,22,23,24]. The chronic misalignment between internal physiological signals and externally imposed timing of day-to-day work increase the risk of mortality for the evening CT [13,20,25,26]. Dietary patterns and food intake exhibit daytime rhythms, with possible influence of CT [1,22,27,28]. Individuals with an evening CT have unfavorable eating habits, tending to consume fewer but larger meals and to skip breakfast and delay food intake due to late awakening [29,30,31].

In Brazil, most people with CNCDs depend on the public health system (PHS) to receive adequate treatment, through the Family Health Strategy (FHS). Brazil is an upper-middle-income economy [32] but still has high inequalities and significant disparities among different regions. In this context, the FHS represents the main policy to favor health accessibility to the population. It plays an important role in reducing social and health disparities in populations of socioeconomic vulnerability [33,34].

Social conditions may influence the period of exposure to light [1] due to work routines and consequently influence CT. Understanding such differences can guide the development of cost-effective collective health promotion strategies. The implication of such chronotype-driven differences in dietetic nutrition is yet to be fully elucidated in people assisted by a PHS. Thus, the present study aimed to investigate the relationship between CT, diet and CNCD, analyzing clinical, sociobehavioral and nutritional aspects.

## 2. Materials and Methods

The present study was approved by the Human Research Ethics Committee of the Federal University of Lavras (COEP/UFLA, MG, Brazil—CAAE: 29523220.0.0000.5148). All procedures were in accordance with the Ministry of Health Resolution 466/12 from the Brazilian Government. According to the protocol, all participants were informed about the research objectives and signed the Free and Informed Consent Form (FICF). The study was carried out in the units of the Family Health Strategy (FHS) in the city of Lavras (approximately 102,000 citizens), latitude 21°14′43 south and longitude 44°59′59 west, in the state of Minas Gerais, Brazil. In accordance with the last survey in 2010, the Human Development Index (HDI) of Lavras corresponds to 0.782, occupying 5th place in the state of Minas Gerais.

### 2.1. Study Design and Sampling

All 17 health units of the city were included in the study, and the participants were selected proportionally and systematically among them. Participants were randomly selected by means of probabilistic sampling by clusters within people registered in the FHS at the time of the project’s start (52,628 individuals). The inclusion criteria comprised individuals from both sexes and over 18 years of age. Pregnant women and people with mental illness were excluded. Individuals with drastic and recent changes in dietary patterns, such as chronic renal patients and athletes, and those under restrictive diets for weight loss/gain were also excluded in order to avoid bias.

For the sample calculation, we considered the estimated prevalence of overweight/obesity/metabolic syndrome in Brazil (~40%) [35,36,37], 95% confidence interval and 5% accuracy, generating a minimum sample of 172 individuals. Due to the great variability in the reported prevalence, this number was increased by 100%, resulting in a minimum number of 344 participants (Figure 1).

Data collection was carried out through cross-sectional home visits, assessing nutritional aspects and clinical and sociodemographic characteristics in order to verify the existence of an association between them and the sample’s CT. On the day of the visit, the Food Frequency Questionnaire (FFQ) [38] was applied and followed by a general anamnesis, anthropometric measurements and CT profile assessment (further description). The information on food was collected with aid of a photo album, and the FFQ was adapted to portion sizes [39]. All procedures were performed by a previously trained nutritionist, except the evaluation of the CT, which was applied by another health professional from the team.

The dietary record was evaluated for the intake of the following nutrients as described previously [40]: omega-3, omega-6, fiber, calcium (Ca), magnesium (Mg), manganese (Mn), phosphorus (P), iron (Fe), sodium (Na), potassium (K), copper (Cu), zinc (Zn), vitamin A (retinol), thiamine, riboflavin, pyridoxine, niacin and vitamin C. In addition to these, the intake of macronutrients such as carbohydrates, proteins and lipids was quantified and associated with the respective energy content. For the estimation of dietary consumption of nutrients, we used a conversion table (TACO table) [41].

Anthropometric measurements were performed according to the Technical Standard of the Food and Nutrition Surveillance System (SISVAN) [42]. We evaluated body weight, height and waist circumference. The body mass index (BMI) was calculated using the formula BMI = (weight (kg))/(height (m))^2^ [43]. Waist circumference was measured and classified to assess the risk of metabolic complications.

We also used the Morningness–Eveningness Questionnaire (MEQ) by Horne and Östberg [5], translated and adapted to Portuguese language [44] to determine each participant’s CT. The interview consists of 19 questions with objective answers previously defined and is self-assessed. For each answer, a specific score is generated, and the total sum of the answers varies from 16 to 86 points. Scores less than 41 points are compatible with “evening types”, between 42 and 58 points are “intermediate types” and higher than 59 points indicate “morning types”.

At the same time, individuals were asked about general life habits, history of diseases, long-term treatment medications and socioeconomic status. The sociodemographic and clinical characteristics evaluated were age, sex, educational level, income, physical activity frequency, smoking, consumption of alcoholic beverages and the presence of CNCDs (diabetes mellitus, hypertension, hypercholesterolemia, hypertriglyceridemia, hypothyroidism, liver steatosis, cardiopathy, cancer history, depression). Some of these variables were dichotomized, namely educational level ≥8 years of study, smoking yes or no and family income up to two minimum wages or higher (approximately USD 500) [40] (Figure 1).

### 2.2. Statistical Analysis

Statistical analysis was performed using SPSS 26.0 software considering an alpha level of 5% by an applied statistics spec (PMC). Descriptive statistics consisted of means, standard deviations and percentages. Missing random data occurred in only 9 observations (monotone missing pattern), and data imputation was performed predicting the missing data by linear regression. Chi-square test was applied to compare the frequencies of categorical variables among the three chronotypes (morning, evening or intermediate).

In order to summarize the number of nutritional variables, principal component analysis was used to estimate the number of components emerging from the intake of the following micronutrients: omega-6, omega-3, fiber, calcium, magnesium, manganese, phosphorus, iron, sodium, potassium, copper, zinc, retinol, thiamine, riboflavin, pyridoxine, niacin and vitamin C [40]. First, the correlation matrix of the standardized variables was examined, and the number of components to retain was based on eigenvalues, total of explained variance and scree plot examination. As the variables showed moderate correlations, the oblimin rotation was performed. The overall Kaiser–Meyer–Olkin (KMO) measure and Bartlett’s test of sphericity were examined as assumptions of the test [40].

Further, Fisher’s discriminant analysis was used to ascertain which of the clinical, sociobehavioral and nutritional aspects (protein, lipid and carbohydrate intake and micronutrient component loadings generated from principal component analysis) would be significant to discriminate the CT groups of participants. The following assumptions of the test were observed: independence of observations, multivariate normality and homogeneity of variance (Box’s M statistic).

With the aim of explaining the variation in the CT scores, the consumption of micronutrients (corrected to the total energy intake) and other individual and sociodemographic variables were used as explanatory factors in the adjustment of a linear regression model. The following micronutrients were included in the analysis: zinc, retinol, omega-6, omega-3, thiamine, copper, niacin, manganese, sodium, cholesterol, vitamin C, pyridoxine, riboflavin, fiber, calcium, tryptophan, iron, magnesium, potassium and phosphorus, in addition to the variables age, sex and BMI. The backward procedure was used to obtain the final model, after examining the changes in the adjusted R2 and F-values for each independent variable excluded from the model. The assumptions of the test, namely normality, collinearity (VIF and tolerance), independence of errors (Durbin–Watson) and homoscedasticity (residual analysis), were also considered to obtain the best fit.

## 3. Results

Table 1 shows an exploratory analysis of the sample, as well the frequencies of chronic diseases according to each CT. The percentage of females was similar between groups, while the morning CT group showed the higher mean age and the lower percentage of individuals with at least 8 years of schooling. The frequency of heart diseases was different among chronotypes, being higher in the evening CT group, although it is important to consider the small number of individuals of this group.

A principal component analysis (PCA) with oblimin rotation was used to identify micronutrient dietary patterns within the study population; an overall Kaiser–Meyer–Olkin (KMO) measure equal to 0.82 was found, and the Bartlett’s test of sphericity was statistically significant (*p* < 0.0001), indicating that the data were likely factorizable. After oblimin rotation of the components, PCA revealed that the first three components explained 75% of the total variance, as confirmed by the visual inspection of the scree plot. As such, three components met the interpretability criterion and were retained, as observed in Table 2.

For interpretation purposes, by examining Table 2 it can be assumed that for Component 1, the higher the component, the higher the loadings of omega-6 and -3, sodium, zinc, thiamine, pyridoxine and niacin intake; for Component 2, the higher the component, the higher the loadings of fiber, magnesium, manganese, iron, potassium, copper and vitamin C intake; and for Component 3, the higher the component, the higher the loadings of calcium, phosphorus, retinol and riboflavin intake.

Further, Fisher’s discriminant analysis was performed to discriminate the three CT groups from the following variables: age, sex, BMI, physical activity, alcohol consumption and nutritional aspects. Two discriminant functions were obtained (F1 and F2), and 87% of the variance was explained by F1 (*p* = 0.001) (Box’s M statistic: *p* = 0.545). According to F1, the standardized coefficients indicate that the variable ‘age’ is the most important predictor, followed by Component 1, sex and BMI, in discriminating groups, as follows:F1 = −0.20 × BMI + 0.72 × Age − 0.68 × Component1 + 0.12 × Protein + 0.17 × Lipid + 0.10 × Carbohydrate + 0.16 × PhysActivity + 0.25 × Sex + 0.03 × Alcohol
F2 = 0.42 × BMI − 0.11 × Age +0.30 × Component1 − 0.19 × Protein − 0.91 × Lipid +0.52 × Carbohydrate + 0.63 × PhysActivity + 0.13 × Sex + 0.17 × Alcohol

As observed in Figure 2, F1 discriminates better between the morning CT (blue dots) and the other groups, as its mean is reasonably different from the others, while F2 helps to discriminate between evening CT (green dots) and the other two groups. Based on the findings, the morning CT classification results from a higher age, lower Component 1 (that is, omega-6 and -3, sodium, zinc, thiamine, pyridoxine and niacin intake), male sex and lower BMI; on the other hand, lower age, higher BMI and higher Component 1 favor the evening CT classification.

The linear regression model was adjusted with the aim of explaining the variation in CT scores (Table 3). The CT scores were predicted by age and the intake of magnesium, sodium, omega-6, fiber, zinc, retinol and pyridoxine (adj R2 = 14%; *p* < 0.001), meaning that higher scores were related to higher age and lower intake of omega-6, fiber, zinc, retinol and pyridoxine. It is possible to observe that the results of the linear model are in agreement with the results of the functions obtained by discriminant analysis, as they share the same hypothesis and the only difference is the dependent variable: CT scores and CT groups, respectively. The regression model showed good fit, as observed by the parameters of tolerance, VIF, residual analysis and independence of errors (Durbin–Watson).

## 4. Discussion

The findings of the present study revealed associations between CT and sociobehavioral and dietary patterns among individuals assisted by the PHS. Morning and evening individuals showed opposite characteristics regarding age, BMI and the ingestion of Component 1 nutrients. The former group included older male individuals with lower BMI and lower intake of Component 1. Morning individuals presented lower educational level. The evening CT group, on the other hand, was composed mainly of individuals with lower age, higher BMI and also higher Component 1 intake. Moreover, this group showed a greater risk of heart diseases.

In the present study, age was the most important predictor for discriminating the morning CT. Important events favoring morningness begin around the age of 50 [45]. These changes include alterations in the expression of several genes related to the circadian clock [46] and a reduction in the secretion of circulating hormones (especially melatonin and cortisol) [47], leading to decreased sleep [48,49].

Another important factor discriminating CT was sex, with most matutine participants being males. Indeed, men’s and women’s biological rhythms are quite different, with the latter tending to go to bed earlier, to wake up earlier and to prefer morning activities [50,51,52,53]. However, the evaluated sample comprises a group of people with social vulnerability. Most of the interviewed women used to be housewives, while men were responsible for working outside the home. This means that men wake up earlier, often even on weekends, a fact that might be related to social jetlag [6].

Most of the morning CT participants showed fewer years of education in our study. The relationship between CT, intelligence and academic performance has been described in a controversial way in the literature [51,54,55]. Evening individuals are positively related to cognitive ability, yet negatively related to indicators of academic achievement [51] due to social jetlag (desynchronosis caused by misalignment between social and biological clocks) [6,55]. Panev et al. found that evening individuals present a higher level of intelligence, but these advantages disappear when social jetlag is greater than 2 h [55]. Mistimed sleep is able to alter the expression of circadian genes [49,56]. However, it is important to highlight that pressures for working and sleep timing can also interfere with the CT. Then, morning CT is more common in people who work on a daily basis [57]. Our data are in accordance with the Brazilian Institute of Geography and Statistics (IBGE), which reported that despite the advances achieved in the latest assessments, more than 50% of the Brazilian population aged 25 or more had not completed basic and compulsory school education [58]. We were surprised by the small number of evening individuals in the present sample. In a previous study in our country, the prevalence of the evening type was 32%, whereas 54% were intermediate, and 14% were morning type. This previous study investigated 648 individuals between 17 and 49 years of age [59]. However, our sample comprised older individuals and individuals living mostly under socioeconomic vulnerability. The average MEQ score increases linearly with age [60], leading to intermediate and morning types. In other words, the higher the age, the higher the score, which is compatible with intermediate or morning CT. Another reason for the large number of morning chronotypes in our study could be that due to unfavorable socioeconomic conditions, many individuals were pressured to wake up very early and work double or night shifts as a strategy to maintain employment and improve family income. This result highlights the importance of raising education and qualification of young people as a way to combat the significant inequality in the country [58].

We also found that evening individuals were more prone to heart diseases. Circadian clocks are present in different cells of the cardiovascular system [61,62]. Several cardiovascular functions such as blood pressure control, heart rate, endothelial function and thrombus formation can be influenced by circadian rhythm [62,63]. There is evidence that Clock genes are involved in the homeostatic function of the endothelium, since these genes control both thrombomodulin and the plasminogen activator inhibitor-1 (PAI-1) in endothelial cells, essential in balancing coagulation and fibrinolysis processes [64,65]. Based on the aforementioned knowledge, significant circadian misalignment is recognized as a risk factor for the development of cardiovascular diseases [13]. Emerging epidemiological evidence links evening CT with cardiovascular disease [66,67]. These individuals are more predisposed to circadian misalignment due to an asynchrony (social jetlag) between their endogenous biological clocks and the time of social activities, such as food intake, work and sleep–wake cycle, making them vulnerable to cardiometabolic dysfunction [13].

Another important finding was the relationship between evening CT and high BMI. Nocturnal preference and greater social jetlag are associated with increased occurrence of overweight/obesity [56,68] and metabolic syndrome [56]. This suggests a broader involvement of the circadian clock in the pathophysiology of CNCDs [69]. Over the years, especially in the last century in modern industrialized society, there have been significant changes in zeitgebers, culminating in shorter sleep durations and higher availability of food [70]. There is more time available to eat, and people generally feel more tired and stressed, hindering their ability to follow exercise or dietary regimens [70]. Moreover, food availability became more abundant, the intake of high-calorie foods and simple sugars has increased, the frequency of snacks has increased and the timing of snacks has changed to later in the day [70,71].

The central biological clock evolved to synchronize activity, feeding and sleep to diurnal and seasonal changes using hormones and the autonomic nervous system [72]. It is postulated that the thrifty genotype has evolved over millions of years to be expressed during seasons with high availability of food, periods in which there would be an increase in insulin secretion to prioritize the increase in fat deposition to prepare adaptively for the following seasons and the possibility of food scarcity [73]. Thus, in a modern Western environment with constant and easy access to food, as well as the social jetlag induced by dramatic changes in zeitgebers, the thrifty genotype would be permanently activated, leading to obesity and its respective pathological biochemical characteristics [73]. The eating behavior of evening individuals is more frequent and irregular, a fact that directly impacts the weight gain and fat mass verified in this group [71]. In this sense, chrononutritional interventions can represent cost-effective strategies with a direct impact on weight control [74].

With regard to energy intake, we found no differences among the three CTs. Mazri and colleagues conducted a scoping review and found that most studies did not show a significant association between CT and total daily energy intake. However, associations between evening CT and higher carbohydrate or fat consumption were reported [1,4]. Interestingly, a higher intake of carbohydrates/energy after 8:00 p.m. prevailed for evening individuals [4]. Mirkka et al. found that the evening types had a lower intake of macronutrients and energy in the morning when compared to the morning types. However, they presented a higher night intake of sucrose, fat and fatty acids, with no difference in the total daily energy intake [71].

In our sample, evening CT was associated with a higher consumption of Component 1 nutrients, which included omega-3 and omega-6 (a subgroup of fats) and some micronutrients. There are limited publications regarding the effects of PUFAs on circadian parameters [75]. Greco et al. found that the use of DHA attenuates the deleterious effects of palmitate (saturated fat) on the circadian profile expression of Bmal1, suggesting a protective effect of DHA [75]. On the other hand, a reduction in the consumption of saturated fats has been observed in Western countries, a fact confirmed in recent Brazilian research [76,77], as has an increase in the intake of omega-6 [76]. Increased consumption of omega-6 can cause negative consequences for metabolic health due to its recognized proinflammatory, prothrombotic and proadipogenic effects [78]. Such a fact could contribute to the increase in the incidence of obesity and heart disease in the evening CT [79]. Evidence suggests that peripheral clocks located in adipocytes may be involved in the inflammatory state that permeates this tissue [80]. In obesogenic conditions, immune cells are recruited into adipose stores that can further amplify the inflammatory response [81]. In males exposed to circadian misalignment, the increase in inflammation and the reduction in insulin sensitivity were doubled when compared to controls who maintained regular night hours. In addition, in misaligned individuals with similar sleep conditions to those in alignment, there was a significant increase in the levels of ultrasensitive C-reactive protein, a sign of systemic inflammation and a predictor of cardiovascular disease [82].

Regarding the association between CT and micronutrients, there are few publications. In situations of deficiency in group B vitamins, there is evidence of a modest influence of supplementation on the regulation of sleep–wake rhythm modulating neurotransmission through participation in the synthesis of serotonin and melatonin [83]. It is clear that pyridoxine (B6) is involved in the synthesis of serotonin from tryptophan and that niacin (B3) can cause a sparing effect of tryptophan, benefiting the synthesis of serotonin and melatonin. Cyanocobalamin (B12) also contributes to the secretion of melatonin. However, melatonin secretion is influenced by light exposure, and its relationship with sleep quality and CT is not linear [83]. We also found a high intake of some group B vitamins, such as thiamine (B1), pyridoxine and niacin, in Component 1, which was observed in the evening CT. However, it is not possible to state that the high intake of these vitamins seen in Component 1 is directly related to supraphysiological levels, as they are water-soluble vitamins (without storage). No association was found between cyanocobalamin intake and CT in our analysis. Regarding the consumption of thiamine, Shibata et al. found similar results in a study with young Japanese women [84]. In a murine experiment with animals deficient in thiamine, after the normalization of consumption, animals’ daily rhythm of regulating body temperature was restored [85]. However, it is not possible to affirm that the greater consumption of thiamine in the evening CT (lower consumption in the morning CT) reflects necessarily excessive/deficient organic levels or even that this has specific repercussions for these individuals.

Many aspects remain uncertain with appreciating micronutrients in regard to the association between mineral intake and CT [4]. Comparative sources in the literature are very scarce, and we found few previous studies that verified these particularities. We believe that the association between the evening CT and increased sodium consumption may be related to an exaggerated consumption of salt. Imaki et al. assessed the eating patterns of individuals with shorter sleep periods and of individuals with preserved sleep and found that those who do not get enough sleep tend to adopt less healthy eating habits, such as a preference for very salty foods [86]. A significant increase in daily zinc intake was also associated with evening CT in our study. This fact may be related to preferences for certain foods, such as beans, which have a guaranteed place in the daily diet of the Brazilian population. However, neutral [87] and opposite [88,89] results have also been previously reported. We believe that the increased consumption of certain micronutrients and ions is a reflection of a quantitative imbalance in the evening individuals, especially in larger meals after 8:00 p.m., since they usually stay up late and eat more meals in irregular schedules and quantities [71] when compared to morning individuals.

Finally, this study was carried in a population assisted by a PHS, configured as the main primary care initiative in Brazil. Although this program has a national coverage of about 70% [90], Brazil is still experiencing a significant increase in the incidence of CNCDs. CNCDs mainly affect people with low income and low education, either due to risk factor exposure or to low access to health information and services [91]. The nutritionist is not part of the FHS basic team. However, they take part in the Family Health Support Nucleus (FHSN), which is an itinerant additional team that assists the basic units (which is not present in all cities). As part of the multidisciplinary team, these professionals may help to establish a link between the population’s nutritional needs and the health service.

Recently, chrononutrition is becoming a prominent field of research encompassing three dimensions of eating behavior: time, frequency and regularity [1]. Considering that the time and nutrient composition of the diet can modulate the biological clock, the identification of CT may represent an interesting tool in the future to manipulate aspects of the diet in public services in order to achieve better clinical responses [92,93].

## 5. Conclusions

In this study, most morning CT individuals were elderly thin males with lower consumption of omega-6 and -3, sodium, zinc, thiamine, pyridoxine and niacin, whereas evening individuals were younger, had higher BMI and had higher consumption of the studied micronutrients. The identification of circadian and behavioral risk groups can help to provide preventive and multidisciplinary health promotion measures.

## Figures and Tables

**Figure 1 nutrients-13-02260-f001:**
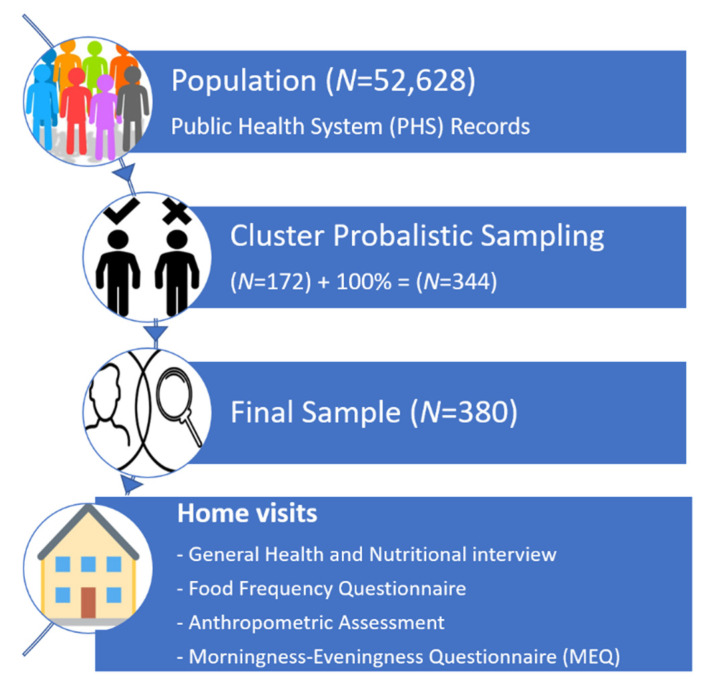
Fluxogram of study design. Creative Commons (CC) Licensing: “Association-community-group-meeting-152746” and “Search-people-selection-find-well-2831336” by OpenClipart-Vectors/27400 images and PaliGraficas, respectively, via Pixabay. Licensed under free for commercial use, no attribution required. “People selection” by B. Farias, CL, via thenounproject.com is licensed under CC 0. “House Emoji Icon” by “Twitter Emoji Follow” via iconscout.com is licensed under CC 0.

**Figure 2 nutrients-13-02260-f002:**
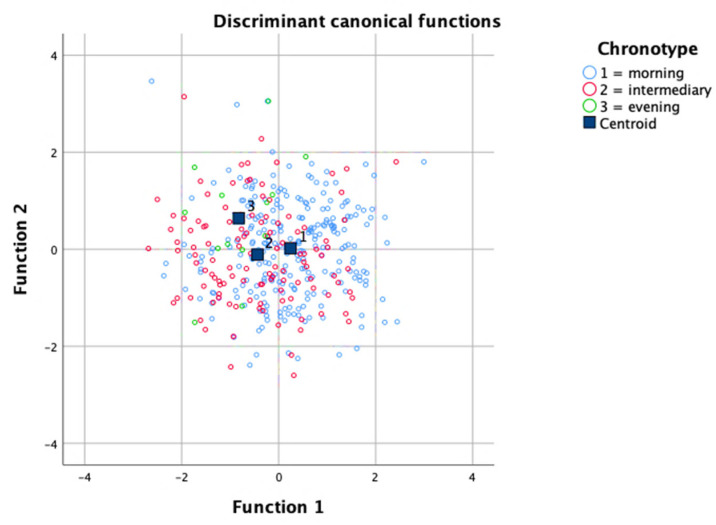
Graphical analysis of the discriminatory power of the two functions and centroids of the three groups. Function 1 (X axis) discriminates better between the morning chronotype (CT) (blue dots) and the other groups, as its mean is reasonably different from the others, while Function 2 (Y axis) helps to discriminate between evening CT (green dots) and the other two groups. Function 1 explained 87% of the variance (*p* = 0.001) (Box’s M statistic: *p* = 0.545).

**Table 1 nutrients-13-02260-t001:** Sociodemographic and clinical characteristics of the sample according to sex.

Chronotype		Morning (*n* = 252)	Intermediate (*n* = 119)	Evening (*n* = 13)
Clinical and sociobehavioral aspects				
Age (years) *	mean (SD)	53.5 (0.7)	46.5 (1.2)	42.4 (2.7)
BMI (Kg/m^2^)	mean (SD)	29.0 (0.4)	29.3 (0.6)	31.4 (2.5)
Sex (female)	%	77	83	85
Schooling * (>8 years)	%	49	66	62
Income (>2 min wage)	%	36	36	9
Physical activity (>3x a week)	%	36.5	24	38.5
Smoking habit (yes)	%	21	21	31
Alcohol consumption (≥2 times/week)	%	10	12	15
Chronic diseases				
Diabetes mellitus (yes)	%	32	24	31
Hypertension (yes)	%	54	52	31
Hypercholesterolemia (yes)	%	26	22	7
Hypertriglyceridemia (yes)	%	2	2.5	0
Hypothyroidism (yes)	%	5	9	0
Liver steatosis (yes)	%	1	2.5	8
Heart diseases (yes) *	%	7.5	1	15
Kidney disease (yes)	%	0.5	1	0
Circulatory system disease (yes)	%	0.5	0	0
Depression (yes)	%	6	2.5	15
Nutritional aspects				
Total energy intake (Kcal)	mean (SD)	1522.3 (27.8)	1633.6 (42.4)	1649.0 (91.5)
Protein intake (g)	mean (SD)	62.3 (1.3)	66.8 (1.9)	63.9 (3.9)
Lipid intake (g)	mean (SD)	42.0 (1.0)	46.1 (1.4)	42.1 (3.6)
Carbohydrate intake (g)	mean (SD)	222.9 (4.1)	235.4 (6.7)	248.2 (16.3)

* *p* < 0.05 (chi-squared test).

**Table 2 nutrients-13-02260-t002:** Micronutrient intake patterns obtained by principal component analysis with oblimin rotation (three components retained).

	Component		
	1	2	3
omega-6	0.784		
omega-3	0.690	0.356	
fiber	0.334	0.833	
calcium			0.883
magnesium		0.722	
manganese	0.319	0.765	
phosphorus	0.343		0.632
iron	0.580	0.599	
sodium	0.720		
potassium		0.656	0.361
copper		0.732	
zinc	0.631		
retinol			0.900
thiamine	0.577		
riboflavin			0.829
pyridoxine	0.719		
niacin	0.716		
vitamin C	−0.389	0.718	

Coefficients smaller than 0.30 are omitted.

**Table 3 nutrients-13-02260-t003:** Final predictive model for estimation of the CT scores.

Dependent Variable	Independent Variables	B	CI (95%)	t	Sig	F	Adj R2	Durbin–Watson
	constant	59.73	50.42–69.04	12.62	<0.0001	8.471	0.138	1.959
CT scores	age	0.20	0.12–0.27	5.01	<0.0001			
	omega-6	−1001.14	−1944.95–−57.33	−2.09	0.038			
	fiber	−431.96	−743.55–−120.37	−2.73	0.007			
	magnesium	69.85	25.99–113.71	3.13	0.002			
	sodium	6.37	0.82–11.92	2.26	0.025			
	zinc	−1337.11	−2397.76–−276.47	−2.48	0.014			
	retinol	−25.93	−42.51–−9.35	−3.08	0.002			
	pyridoxine	−8509.37	−14,441.69–−2577.06	−2.82	0.005			

CT: Chronotype; B: Beta coefficient; CI: confidence interval; t: t statistics; F: F statistics; Adj R2: adjusted R squared

## Data Availability

Data will be available upon request.

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
