# Peer review of "Association between Chronotype and Nutritional, Clinical and Sociobehavioral Characteristics of Adults Assisted by a Public Health Care System in Brazil"

_nutrients, 2021, doi:10.3390/nu13072260_

Round 1
Reviewer 1 Report
This manuscript is interesting and noble study to show that influence of chronotype on nutritional, clinical and socio-behavioral aspects under the Brazilian public health care systems. Overall, it is beneficial to publish in 'Nutrients' journal.
However, there are some points to improve:
- Figure resolution is not clear. Please use high-resolution format or I strongly recommend re-drawing figures with a special program such as Photoshop-Illustrator.
- Figure 1 legend seems not organized. Please re-write the explain of the figure.
- In Figure 2, authors mentioned color but the figure is black-white. Please insert color-figure or correct the manuscript to understand the differences among CTs. And it needs more detailed legend for understand.
- Evening CT study samples are too low (n=13) compared to Morning or Intermediate ones. I am not sure if it is just simple typo (103 or 130) or really less sample numbers. If possible, please do your best to increase the No. of samples. It has to make your paper strong and solid. Or, the authors need to discuss the reason why Eventing CT persons are less than other CT persons.
- Even though the manuscript shows valuable sight for public health and social study methods, we should be careful to conclude based on experiment results. When there is no statistical significance, please clarify it; ex) it has a trend.... When you say that there is a relation between A and B, you should present statistical justification.
Author Response
Reviewer 1
This manuscript is interesting and noble study to show that influence of chronotype on nutritional, clinical and socio-behavioral aspects under the Brazilian public health care systems. Overall, it is beneficial to publish in 'Nutrients' journal.
However, there are some points to improve:
- Figure resolution is not clear. Please use high-resolution format or I strongly recommend re-drawing figures with a special program such as Photoshop-Illustrator.
Answer: We improved figure resolution following reviewer’s suggestion.
- Figure 1 legend seems not organized. Please re-write the explain of the figure.
Answer: Indeed, during the edition of the final file something went wrong. We organized figure legend as suggested by the reviewer. Page 4. Lines 138-151.
- In Figure 2, authors mentioned color but the figure is black-white. Please insert color-figure or correct the manuscript to understand the differences among CTs. And it needs more detailed legend for understand.
Answer: We replaced that B&W figure by a color one. Besides, a more detailed description of the figure was included in the legend. Page 7 Lines 234-238
- Evening CT study samples are too low (n=13) compared to Morning or Intermediate ones. I am not sure if it is just simple typo (103 or 130) or really less sample numbers. If possible, please do your best to increase the No. of samples. It has to make your paper strong and solid. Or, the authors need to discuss the reason why Eventing CT persons are less than other CT persons.
Answer: This was not a typo. The real number was indeed 13. The chronotype distribution was not intentional. It just happened at random.
An additional discussion was included to justify the present findings: (Page 8. Lines 286-297).
We were surprised by the small number of evening individuals in the present sample. In a previous research in our country, the prevalence of evening-type was 32%, whereas 54% were intermediate, and 14% morning-types. This previous study investigated 648 individuals between 17-49 years [60]. However, our sample comprised older individuals and living mostly under socioeconomic vulnerability. The average MEQ score increases linearly with age [61], leading to intermediate and morning types. In other words, the higher the age, the higher the score, which is compatible to intermediate or morning CT. Besides to age, we believe that, due to unfavorable socioeconomic conditions, many individuals were pressured to wake up very early, work double or night shifts as a strategy to maintain employment and improve family income.
- Alam MF, Tomasi E, Lima MS, Areas R, Menna-Barreto L. Characterization and distribution of chronotypes in southern Brazil: gender and season of birth differences. J Bras Psiquiatr. 2008; 57(2). https://doi.org/10.1590/S0047-20852008000200001
- von Schantz M, Taporoski TP, Horimoto AR, Duarte NE, Vallada H, Krieger JE, Pedrazzoli M, Negrão AB, Pereira AC. Distribution and heritability of diurnal preference (chronotype) in a rural Brazilian family-based cohort, the Baependi study. Sci Rep. 2015 Mar 18;5:9214. doi: 10.1038/srep09214.
- Even though the manuscript shows valuable sight for public health and social study methods, we should be careful to conclude based on experiment results. When there is no statistical significance, please clarify it; ex) it has a trend.... When you say that there is a relation between A and B, you should present statistical justification.
Answer: We agree with the reviewer and conclusions were revised accordingly. Page 11 Lines 414-418
“It was concluded that most morning chronotype individuals were elderly thin males with lower consumption of Omega-6 and 3, Na, Zn, Thiamine, Pyridoxine and Niacin, whereas evening individuals were younger, with higher BMI, and with higher consumption of the referred micronutrients. The identification of circadian and behavioral risk groups can help to provide preventive and multidisciplinary health promotion measures.”
Reviewer 2 Report
The study titled "Influence of Chronotype on Nutritional, Clinical and Socio-behavioral Characteristics of Adults Assisted by a Public Health Care System in Brazil" aimed to investigate the relationship between chronotype, diet and chronic non-communicable diseases, analyzing clinical, socio-behavioral, and nutritional aspects.
The cross-sectional design of study doesn't let the authors conclude cause and effect relationship. Then, the title of article is misleading. The statistical analysis applied in the current study is very limited. Further details on descriptive analyses are needed. Multivariable nonlinear and linear regression models can be applied to further explore the findings. There are several continuous variables in the study such as chronotype and BMI. The non-linear relationships of these variables have been demonstrated in the literature.
Author Response
- The study titled "Influence of Chronotype on Nutritional, Clinical and Socio-behavioral Characteristics of Adults Assisted by a Public Health Care System in Brazil" aimed to investigate the relationship between chronotype, diet and chronic non-communicable diseases, analyzing clinical, socio-behavioral, and nutritional aspects. The cross-sectional design of study doesn't let the authors conclude cause and effect relationship. Then, the title of article is misleading. The statistical analysis applied in the current study is very limited. Further details on descriptive analyses are needed. Multivariable nonlinear and linear regression models can be applied to further explore the findings. There are several continuous variables in the study such as chronotype and BMI. The non-linear relationships of these variables have been demonstrated in the literature.
Answer: We agree with the reviewer that cross-sectional studies do not infer cause-effect relationship. It was not our intention to infer that. In that sense, we changed the word “Influence” in the title for “Association” to avoid this misunderstanding.
As suggested, an additional analysis was provided with the aim at obtaining a predictive model for chronotype scores using the independent variables under study (Table 3). It is possible to observe that the results of the linear model are in agreement with the results of the functions obtained by discriminant analysis, as they share the same hypothesis and the only difference is the dependent variable: chronotype scores and chronotype groups, respectively. As the reviewer indicated that both results may be interesting for the reader, the linear regression model was included in the Results section.
Round 2
Reviewer 2 Report
The authors applied part of my previous comments. Given the evidence in literature review, application of nonlinear regression methods such generalized additive model is a must. Without that, the conclusion could be misleading.
Author Response
Dear editors and reviewer 2,
Please find attached the response letter to reviewers' queries.
Kind regards
